# A Method for In Situ Reverse Genetic Analysis of Proteins Involved mtDNA Replication

**DOI:** 10.3390/cells11142168

**Published:** 2022-07-11

**Authors:** Natalya Kozhukhar, Domenico Spadafora, Yelitza A. R. Rodriguez, Mikhail F. Alexeyev

**Affiliations:** Department of Physiology and Cell Biology, University of South Alabama, Mobile, AL 36688, USA; nkozhukhar@southalabama.edu (N.K.); dspadafora@southalabama.edu (D.S.); rodriguez@southalabama.edu (Y.A.R.R.)

**Keywords:** GeneSwap approach, mtDNA replication, mtDNA metabolism, mtDNA transcription, mtDNA instability, TFAM, TFAM chimeras, TFAM knockout, TFAM-mtDNA evolutionary co-adaptation

## Abstract

The unavailability of tractable reverse genetic analysis approaches represents an obstacle to a better understanding of mitochondrial DNA replication. Here, we used CRISPR-Cas9 mediated gene editing to establish the conditional viability of knockouts in the key proteins involved in mtDNA replication. This observation prompted us to develop a set of tools for reverse genetic analysis in situ, which we called the GeneSwap approach. The technique was validated by identifying 730 amino acid (aa) substitutions in the mature human TFAM that are conditionally permissive for mtDNA replication. We established that HMG domains of TFAM are functionally independent, which opens opportunities for engineering chimeric TFAMs with customized properties for studies on mtDNA replication, mitochondrial transcription, and respiratory chain function. Finally, we present evidence that the HMG2 domain plays the leading role in TFAM species-specificity, thus indicating a potential pathway for TFAM-mtDNA evolutionary co-adaptations.

## 1. Introduction

In most mammalian cells, mitochondria generate the bulk of ATP through the process of oxidative phosphorylation (OXPHOS). Mitochondrial DNA (mtDNA) encodes critical components of four out of five OXPHOS complexes, and therefore mtDNA maintenance, transcription, and translation of mitochondrial transcripts are central to cellular bioenergetics. Alterations in mtDNA maintenance and gene expression have been linked to mitochondrial diseases [1,2], cancer [3], diabetes [4], cardiovascular disease [5], and neurodegenerative disorders [6], as well as the normal process of aging [7]. Understanding the mechanisms of mtDNA maintenance is therefore of utmost importance as it can identify targets for clinical interventions aimed at the prevention and treatment of disease.

Reverse genetic analysis, made possible by recombinant DNA technology, involves altering the protein sequence and subsequent functional assessment of the resulting mutant(s). However, techniques that would allow to routinely extend such analysis to the critical proteins involved in mtDNA replication are not yet available. At least in part, this is due to the fact that whole-body inactivation of these proteins in experimental animals has proven lethal, and the cultivation of cells from animals or embryos knocked out (KO) for genes encoding these proteins has not been reported, leading to the notion that inactivation of these genes may result in lethality through growth arrest [8]. In situ screens also indicated that these genes are essential in most settings [9,10,11]. As a result, attempts to cultivate KO cells for any critical component of the mtDNA replication apparatus have been unsuccessful until recently. However, while this work was in progress, Kang’s group succeeded in inactivating *POLRMT, TFB2M*, and *POLG2* in cultured cells [12,13].

TFAM is arguably the best understood key component of mitochondrial transcription and replication machineries. It is a member of the HMGB subfamily of a high mobility group (HMG) DNA-binding proteins, which are involved in various functions, including DNA repair, immune responses, and wound healing [14]. TFAM consists of five distinct domains: a cleavable Matrix Targeting Sequence (MTS), two HMG domains connected by a linker, and a tail (Figure 1). The human and murine TFAMs also contain a short leader sequence located between MTS and HMG1. Whole-body *TFAM* knockout (KO) is embryonically lethal and is accompanied by severe mtDNA depletion [15]. In contrast, tissue-specific *TFAM* KOs have variable phenotypes, some of which are relatively mild [16,17,18].

Apart from its canonical mitochondrial localization, TFAM has also been detected in the nucleus, where it was suggested as playing a regulatory role [19,20]. This nuclear role is supported by cloning a dedicated, alternatively-spliced nuclear isoform of chicken TFAM [8]. Since the cultivation of TFAM KO cells has not been reported, it remains unresolved whether it is the loss of the nuclear or mitochondrial TFAM function (or both) that mediate the reported lethality associated with the loss of this protein in whole animals and cultured cells [8,15].

The inability to culture cells with complete TFAM ablation [8,21,22,23,24] has impeded a more comprehensive understanding of this protein’s function in mtDNA replication, and the limited available evidence is derived predominantly from observations made in cells that co-express both wild type (wt) and altered forms of TFAM [8,25]. In these experiments, co-expression of the wt protein confounds interpretation and makes impossible unambiguous attribution of a phenotype to a specific TFAM mutation. Therefore, the only reliable information regarding the effects of mutations in human TFAM (hTFAM) on human mtDNA (hmtDNA) replication in situ available to date is derived from two pedigrees with mtDNA depletion [26,27]. Here, we developed a GeneSwap approach that circumvents many of these limitations.

## 2. Materials and Methods

### 2.1. Cell Growth and Treatment

143B cells were from ATCC (Manassas, VA, USA, Cat# CRL-8303). All cells were propagated in Dulbecco’s Modified Eagle Medium (DMEM) containing 10% Fetal Bovine Serum, 50 µg/mL gentamycin, 50 µg/mL uridine, and 1 mM sodium pyruvate in a humidified atmosphere containing 5% CO_2_ at 37 °C. This medium is permissive for the growth of cells devoid of mtDNA (ρ^0^ cells; +UP medium). When indicated, uridine and pyruvate were omitted from this medium for selection against ρ^0^ cells (−UP medium).

### 2.2. Recombinant DNA

Plasmids and viral constructs were generated using standard techniques [28], and diagrams of the core constructs are presented in Appendix A. TFAM chimeras were generated by overlap extension PCR [29]. Crossover points between human and variant TFAMs were at the junction between the linker domain and HMG2 (a vertical arrow in Figure 1A).

### 2.3. GeneSwap

The GeneSwap approach is implemented in GeneSwap cell lines. The outline of engineering a GeneSwap cell line is presented in Figure 1 using 143B *TFAM* GeneSwap cell line 143B#6 as an example.

To generate pMA3965 plasmid that encodes sgRNA directed at the second exon of the human *TFAM* gene, oligos accgAGGTGGTTTTCATCTGTCT and aaacAGACAGATGAAAACCACCT were annealed and cloned into BsaI-digested plasmid pMA3735 (Appendix A). To inactivate *TFAM,* 143B cells were transfected with a mixture of 0.7 μg pMA3965, 0.3 μg of pAcGFP1-1 plasmid (Clontech, Cat# 632497), and 2 ug of pX330 (Addgene, Cat# 42230). The purpose of pAcGFP1-1 plasmid is to label transfected cells for flow cytometry. Therefore, any EGFP or other fluorescent protein-encoding plasmid can be used instead of pAcGFP1-1. Transfected (AcGFP-positive) cells were flow-sorted using BD FACSAria and plated into DMEM + UP medium in 150-mm dishes at 200, 400, and 1000 cells/dish. After the appearance of colonies, they were picked into 24-well plates with DMEM + UP, expanded, and analyzed for the presence of mtDNA using DirectPCR reagent (Viagen Cat# 301-C) according to the manufacturer’s recommendations using primers listed in Appendix A (Figure 1D).

To confirm inactivation of both *TFAM* alleles, a 601bp DNA fragment encompassing the targeted region was amplified from ρ^0^ clones with primers GAGCTGGAGTATAGACGCTTTC and CTTCCTAGGGTGCTTTCTACAC. The PCR fragment was cloned in EcoRV-digested pBluescriptII SK+ plasmid and inserts from 10 independent clones were sequenced by Sanger sequencing (Appendix A). The lack of TFAM expression was also confirmed by Western blotting (Figure 1E). One clone (#8) was chosen for further modification.

To rescue TFAM deficiency in clone #8, wt *hTFAM* was flanked by loxP sites by cloning *hTFAM* cDNA into the polylinker of rv.3998 (Appendix A). Then, clone#8 was transduced with the resulting rv.4000 (Appendix A).

To reintroduce mtDNA into the resulting rescued cells, they were fused with chemically enucleated MDA-MB-231 cells. MDA-MB-231 cells were chosen as a donor to enable differentiation, by means of STR profiling of true cybrids from donor cells that escaped mitotic inactivation. Briefly, MDA-MB-231 cells were treated for 2 h with 10 μg/mL mitomycin C in DMEM medium to mitotically inactivate them. These chemically enucleated cells (10^6^) were co-plated with 2 × 10^6^ rv.4000-rescued clone#8 cells into a 35-mm tissue culture dish, allowed to attach for 2 h. Then, cells were fused by exposing them for 1 min to 1 mL of the sterile solution made by mixing 4.7 g polyethyleneglycol-1450 (Millipore Sigma, Cat# P5402, Sigma-Aldrich, St. Louis, MO, USA), 4 mL unsupplemented DMEM and 1 mL DMSO (Thermo Fisher Scientific, Waltham, MA, USA, Cat#BP231-100). After fusion, cells were washed three times with unsupplemented DMEM, incubated overnight in DMEM + UP, trypsinized, and 10%, 30%, or 60% of the resulting cell suspension were plated in 150-mm dishes in DMEM-UP to select for cybrids.

To validate cybrids, they were transduced with rv.3491 [30], which encodes Cre recombinase and G418 resistance (Appendix A). Serial dilutions of transduced cells were plated in DMEM + UP supplemented with 800 μg/mL G418 (GoldBio, Cat#G-418-5, St. Louis, MO, USA). The resulting clones were picked into a 24-well plate, expanded, and tested for the presence of mtDNA using DirectPCR reagent as above and for wt *hTFAM* excision using primers listed in Appendix A. Those clones which underwent wt *hTFAM* excision also lost their mtDNA (Figure 1C,G–I Cybrid/Cre). One of these clones was designated 143B#6 and used for further validation.

To further validate the 143B#6 clone, rv.5460 was generated by cloning wt *hTFAM* between PhiC31 attP and attB sites in rv.4659 (Appendix A). To implement the GeneSwap, 143B#6 cells were cotransduced with rv.3442 and rv.5460. As this cotransduction is anticipated to result in the simultaneous excision of the wt *hTFAM* encoded by rv.4000 and introduction of wt *hTFAM* encoded by rv.5460, the resulting cotransductants were expected to retain their mtDNA. Indeed, clones resistant to both puromycin and G418 retained their mtDNA (Figure 1I, GeneSwap). One of these clones was further validated by excising *hTFAM* with PhiC31 recombinase encoded by the retrovirus rv.5136 (Appendix A). Hygromycin-resistant clones from this transduction were picked in 24-well plate, expanded, and tested by PCR for hTFAM excision and mtDNA retention. All clones that underwent PhiC31-mediated excision lost their mtDNA (Figure 1J, GeneSwap/PhiC31).

### 2.4. mtDNA Diagnostics

The presence of mtDNA in human cells was established by duplex PCR with two pairs of primers, one specific to nDNA and another specific for mtDNA as described previously [31], using primers listed in the Appendix A.

### 2.5. Production of Retroviruses

Phoenix Ampho cells (ATCC CRL-3213) were plated in 60-mm dishes overnight at 4 × 10^5^ and 6 × 10^5^ cells per plate to achieve ~70% confluency the next morning. Cells were transfected with a mixture of 5 μg of retroviral vector plasmid plus 5 μg of GAG-Pol plasmid using linear polyethyleneimine (PEI) MW = 25,000 (Polysciences, Cat#23966-100, Warrington, PA, USA) at 1:3 (*w*:*w*) ratio of DNA to PEI. Retrovirus-containing supernatants were collected 48 h after transfection, filtered through 0.45 μm filters (Fisher Cat# 09-928-063, Thermofisher, Waltham, MA, USA), and stored at −80 °C.

### 2.6. Retroviral Transduction

Recipient cells were seeded at 20–40% confluence in wells of 6-well plates and allowed to attach for 4–24 h. Once cells were attached, the medium was replaced with 2 mL of a mixture consisting of 1 mL fresh complete DMEM plus 1 mL of retroviral supernatant(s) supplemented with 10 µg/mL polybrene (Millipore Sigma, Cat#H9268). After overnight incubation in a CO_2_ incubator, the medium was replaced with 2 mL of fresh DMEM medium, and cells were incubated in this medium for another 24 h, after which they were dissociated with 0.05% trypsin (Fisher Cat# MT25052CV) and 10%, 1%, and 0.1% of total cell mix were plated into 150-mm dishes in DMEM + UP medium supplemented with antibiotics specified by retrovirally-encoded resistance genes. After the appearance of colonies, they were picked into 24-well plates, expanded, and analyzed by PCR using DirectPCR reagent (Viagen Cat# 301-C, Austin, TX, USA) according to the manufacturer’s recommendations.

### 2.7. PhiC31-Mediated Excision of Proviral Inserts

TFAM variants were excised by transducing cells with rv.5136 (Appendix A), which encodes a codon-optimized PhiC31 recombinase (PhiC31_o_) and hygromycin resistance. In a few cases, excision was achieved by transiently transfecting cells with pMA4854, which encodes both EGFP and PhiC31_o_. In those cases, EGFP-positive cells were FACS-sorted and plated in +UP media. Successful excision was verified by PCR genotyping using primers listed in Appendix A.

### 2.8. Genotyping of hTFAM Excision in 143B#6 Cells

PCR diagnostics of *hTFAM* excision in 143B#6 cells was conducted using primers listed in Appendix A according to the scheme presented in Figure 1G.

### 2.9. Cellular Respiration

Cellular respiration was evaluated with an XFe24 extracellular flux analyzer (Agilent, Billerica, MA, USA) as recommended by the manufacturer. Briefly, 25,000 cells per well were plated in DMEM +UP medium and allowed to attach overnight. The next day, the medium was exchanged for the assay medium, cells were adapted for 1 h to the assay medium at 37 °C without CO_2_, and a built-in real-time ATP assay protocol was run using Wave software. The results were normalized to protein, which was determined using BCA protein assay kit (Thermo Fisher Scientific, Cat# PI23225).

### 2.10. Western Blotting

Western blotting was performed as described previously [30]. Antibodies used were: anti-MT-CO1 and anti-MT-CO2 (Abcam, Cambridge, MA, USA, Cat# ab14705 and ab91317, RRID:AB_2084810 and AB_10712683, respectively); anti-hTFAM, N-terminal (Cell Signaling Technology Cat# 7495, RRID:AB_10841294); anti-hTFAM, C-terminal (Santa Cruz Biotechnology Cat# sc-376672, RRID:AB_11150497); anti m + h TFAM (PhosphoSolutions Cat# 2001-TFAM, RRID:AB_2492259); anti-TFB2M (Proteintech Cat# 24411-1-AP, RRID:AB_2879530), anti-PolRmt (Abcam Cat# ab32988, RRID:AB_873619); anti-SSBP1 (Proteintech Cat#12212-1-AP, RRID:AB_2195320), anti-PolG1 (Cell Signaling Technology Cat# 13609, RRID:AB_2750886); anti-β-actin (Sigma-Aldrich Cat# A5441, RRID:AB_476744), anti-α-actinin (Santa Cruz Biotechnology Cat# sc-17829, RRID:AB_626633); Goat Anti-Mouse IgG (H + L), HRP Conjugate (Boster Biological Technology, Cat# BA1050, RRID:AB_2904507); Goat-anti-rabbit IgG (H + L), HRP conjugate (Advansta, Cat# R-05072-500, RRID:AB_10719218).

### 2.11. Amino Acid Alignments

Amino acid alignments and percentages of similarity/identity were derived using AlignX algorithm of the VectorNTI package (Thermo Fisher Scientific, Waltham, MA, USA).

### 2.12. Testing oTFAM MTSs

*oTFAM* MTSs were amplified by PCR and fused in-frame with *EGFP*. The fusion constructs were placed under the control of the CMV promoter. The resulting plasmids were transiently transfected into 143B cells using linear polyethyleneimine and imaged using Nikon A1R confocal system after staining mitochondria with MitoTracker Red CMXRos (Thermo Fisher Scientific, Waltham, MA, USA, Cat#M7512).

### 2.13. Growth Rates

Growth rates were determined by plating cells into 6-well plates, allowing them to attach overnight, and determining cell counts in triplicate wells using Beckman–Coulter Z1 particle counter the next day after seeding and again 96 h later. The results were presented as log_2_ (fold change in cell number).

### 2.14. Quantitation of Mitochondrial Transcripts

Quantification of mitochondrial transcript was performed by RT-qPCR using primers listed in Appendix A. RNA was isolated using EZ-10 DNAaway RNA Miniprep Kit (Bio Basic, Amherst, NY, USA, Cat# BS88136) and treated with gDNA removal kit (Enzo Life Sciences, Farmingdale, NY, USA, Cat# ENZ-KIT136-0050) to reduce mtDNA contamination prior to reverse transcription with SensiFast cDNA synthesis kit (Bioline USA, Taunton, MA, USA, Cat# BIO-65053), which was supplemented with primers for *MT-ND6* RT-qPCR. In most experiments, three transcripts representative of three mitochondrial promoters were quantitated using SYBR Fast kit (Roche Holdings AG, Basel, Switzerland, Cat# KK4601): *MT-ND6* (for the light strand promoter, LSP), *MT-RNR2* (for the heavy strand promoter 1, HSP1), and *MT-ND1* (for the heavy strand promoter 2, HSP2). Some experiments also included quantitation of *MT-CO1* transcripts.

### 2.15. Gene Inactivation by CRISPR-Cas9

gRNAs were designed using the CCTop software [32]. To target *TFAM, POLRMT, TFB2M, POLG1, POLG2*, and *SSBP1* loci, we used gRNAs AGGUGGUUUUCAUCUGUCU, GUUUGAGCCCCGCCGCUCCG, UCCGCCAAGGAAGGCGUCUA, CGGGCCCTGGTGTTCGACG +ATATGGCCACCGCCAATGT, CGCTCTCGTGTAGCCGTCA + AAGUCGCACGCGGAGCTCG, and GUGCACUACUUGGGCGAGU, respectively. Targeting oligonucleotides were cloned under control of the U6 promoter into the pMA3735 vector (Appendix A), which was digested with BsaI to release a DNA fragment encoding the LacZα. The resulting plasmids were cotransfected into 143B osteosarcoma cells along with the pX330 encoding humanized *S.pyogenes* Cas9 (a gift from Feng Zhang (Addgene plasmid # 42230; RRID:Addgene_42230 [33], Watertown, MA, USA), pExodus CMV.Trex2 (a gift from Andrew Scharenberg (Addgene plasmid # 40210; RRID:Addgene_40210 [34]), and an EGFP-encoding plasmid. EGFP-positive cells were separated by Fluorescence-Activated Cell Sorting (FACS) and plated at 300–600 cells per 150 mm dish to form colonies. The resulting colonies were analyzed by PCR for the loss of mtDNA (ρ^0^ phenotype), after which the inactivation of targeted genes was confirmed by PCR-cloning-sequencing of the targeted locus and by Western blotting where antibodies were commercially available.

### 2.16. TFAM Orthologs

cDNAs for human, mouse, and rat *TFAM*s were cloned by RT-PCR. The cDNA for Drosophila *TFAM* was a kind gift of Dr. Joseph Bateman. Human codon-optimized versions of the remaining *TFAM*s were synthesized by Twist Bioscience (South San Francisco, CA, USA) based on the back translation of NCBI protein entries (Table 1 and Appendix A). *oTFAMs*, their variants, and chimeric *TFAM*s were cloned in retroviral vector pMA4659 (Appendix A).

### 2.17. MtDNA Copy Number (mtCN)

MtDNA copy number was determined either Taqman duplex qPCR as described earlier [31] or by direct digital droplet PCR (dddPCR [35]). For dddPCR, cells were collected by trypsinization, counted, ~10^6^-cells pellets were generated and frozen at −80 °C. Pellets were resuspended in PBS at ~10,000 cells/μL, 10 μL aliquots were removed and mixed with 90 μL of solution containing 50 μg proteinase K, 40 ul of H_2_O and 50 ul of the DirectPCR solution (Genprice Inc., San Jose, CA, USA Cat# 388-302-C), the mix was incubated at 50 °C for 30 min and then at 95 °C for another 30 min, the solution was adjusted to 500 μL with H_2_O, and 3 μL of the resulting solution was used as template in 20 μL ddPCR reaction to determine nDNA content using primers and probes listed in the Appendix A. For mtDNA quantification, nDNA samples were diluted 500- fold, and 3 μl of the resulting dilution were used in 20 μLddPCR reactions with primers and probes listed in the Appendix A. ddPCR reactions contained 0.9 μM of each forward and reverse primer, 0.25 μM probe, 10 μL of the 2× ddPCR Supermix for Probes (No dUTP), 10 units of EcoRI HF restriction enzyme (New England Biolabs, Beverly, MA, USA, Cat# R3101S), and the balance of water. The cycling parameters were as follows: initial denaturation for 10 min at 95 °C, followed by 40 cycles of 20 s at 94 °C + 1 min at 60 °C, followed by 10 min at 98 °C, followed by the hold at 4 °C. Each sample was measured in 2 or 3 technical replicas. To calculate mtCN per cell, the concentration of mtDNA targets was multiplied by the dilution factor and divided by 0.5× concentration of nDNA targets. Each mtDNA template concentration was combined with each nDNA template concentration generating either 4 (2 technical replicates) or 9 (3 technical replicates) values for mtCN for each sample.

### 2.18. Statistical Analyses

Statistical analyses were performed using one- or two-way ANOVA with post hoc Tukey or Dunnet corrections, respectively, as indicated in Figure legends with the help of GraphPad Prizm v.9.1.0 software package.

## 3. Results

### 3.1. TFAM, POLRMT, TFB2M, POLG1, POLG2, and SSBP1 Are Dispensable for Viability in Cultured Cells

Where tested, the whole-body inactivation of the key components of mtDNA replication apparatus proved lethal [15,36,37,38,39] and the cultivation of knockout cells from these animals, to our knowledge, has not been reported so far. At least for TFAM, dual nuclear and mitochondrial localization has been reported [8,19,20]. This, along with the fact that cultivation of cells devoid of mtDNA (ρ^0^ cells) is not uncommon, leaves open the possibility that the deprivation of the TFAM essential nuclear function(s) of these proteins, compromises cell viability. Therefore, we first set out to establish whether nuclear TFAM or other key proteins involved in mtDNA replication are essential. To this end, we evaluated the viability of the human osteosarcoma 143B cells after inactivating *TFAM, POLRMT, TFB2M, POLG, POLG2*, or *SSBP1* with CRISPR-Cas9. The inactivation was verified by (a) sequencing the targeted chromosomal loci (Appendix A) and (b) by Western blotting for the targeted protein where antibodies were commercially available. In each instance, KO cells lost their mtDNA and were viable in the +UP medium. Consistent with the previous observations of ρ^0^ cells [30], KO cells were unable to grow in the medium devoid of uridine and pyruvate (Figure 1), which may explain the previously reported lethality of the *TFAM* KO in cultured cells [8].

### 3.2. The GeneSwap Technique

Conditional, rather than absolute, lethality of the KO in the key proteins in mtDNA replication opens opportunities for the reverse genetic analysis of these proteins in situ. It allows for the culturing of cells that express only altered versions of TFAM, including those in which function has been severely compromised or completely lost. To capitalize on this, we implemented what we called the GeneSwap approach (Figure 1) for the analysis of hTFAM. At the core of the GeneSwap approach is the simultaneous introduction of the retrovirally-encoded Cre recombinase for the inactivation of endogenous floxed alleles and a wt or altered version of the gene-of-interest (GOI, in our case, *TFAM*). Simultaneous, rather than sequential, introduction of an altered allele prevents intermediate mtDNA loss and alleviates the need to reintroduce mtDNA. This allows for substantial (3-6 weeks) time savings and eliminates interpretational ambiguities associated with the inability to reintroduce mtDNA into cells that express altered GOI. 

### 3.3. GeneSwap of the hTFAM

To validate the GeneSwap approach in human cells, we established a 143B#6 *hTFAM* GeneSwap human osteosarcoma cell line (Figure 1B). 143B *TFAM* KO ρ^0^ cells were transduced with a retrovirus that encodes a wt *hTFAM* flanked by loxP sites (Figure 1G, KO/Wt^lox^. rv.4000, Appendix A), and mtDNA was reintroduced into transduced cells by fusing them with chemically enucleated MDA-MB-231 cells (Figure 1G, KO/Wt^lox^ = cybrid). One of the resulting cybrid clones (designated 143B#6) was selected for further studies and validated. The validation consisted of three steps. (1) Cells were tested for their ability to undergo excision of the provirus-encoded wt *hTFAM* gene with a concomitant loss of mtDNA in response to transduction with a retrovirus rv.3491 (encodes Cre recombinase, [30]) as seen in Figure 1G, cybrid/Cre. (2) They were also tested for the ability to undergo a GeneSwap upon cotransduction with rv.3442 and rv.5460, which encode Cre recombinase and wt *hTFAM* flanked by attP and attB sites for PhiC31 recombinase, respectively (Appendix A). When cotransduced with rv.3442 + rv.5460, 143B#6 cells are expected to retain mtDNA due to the reintroduction of the wt *hTFAM* (Figure 1G, cybrid/Cre + WT). (3) Finally, the resulting cells were tested for their ability to lose the rv.5460-encoded wt *hTFAM* and mtDNA in response to transduction with retrovirus rv.5136 (Appendix A), which encodes PhiC31 recombinase (Figure 1H, GeneSwap + PhiC31).

### 3.4. The Phylogenetic Relationships Do Not Determine the Ability of TFAM Orthologs to Support hmtDNA Replication

Human cells are unable to replicate murine mtDNA and vice versa. Moreover, attempts to introduce mtDNA from some primates into human cells were unsuccessful [40]. The biochemical basis for this interspecies barrier for mtDNA replication (IBMDR) is currently unknown. However, since TFAM has been demonstrated to bind sequence-specifically upstream of some mitochondrial promoters [41,42], a failure of TFAM to bind to a promoter(s) responsible for the synthesis of mtDNA replication primers and/or stimulate transcription from these promoters provides a plausible mechanistic basis for the existence of IBMDR.

A tissue-specific replacement of mTFAM with *hTFAM* in mouse hearts resulted in viable animals with near-normal mtCN and steady-state levels of mitochondrial transcripts suggesting that *mTFAM* and *hTFAM* are interchangeable in mouse cells [43]. However, it is currently unknown whether the reverse holds true or how far down the evolutionary tree does this mutual interchangeability of TFAMs extend. Therefore, we tested TFAM orthologs (oTFAMs) from different taxonomic groups (Table 1) for their ability to substitute for hTFAM.

Of twenty-nine oTFAMs tested, thirteen failed to support hmtDNA replication (Table 1, Figure 2). Unexpectedly, TFAMs from the marsupial Tasmanian devil (*Sarcophilus harrisii*) and chicken (*Gallus gallus*) were in this group. Conversely, TFAMs from phylogenetically more distant frog (*Xenopus laevis*), zebrafish (*Danio rerio*), and even fossil fish coelacanth (*Latimeria chalumnae*) supported hmtDNA replication. To confirm these observations, we interrogated TFAM from another marsupial, opossum (*Monodelphis domestica*), as well as TFAM from the bald eagle (*Haliaeetus leucocephalus*) and a monotreme platypus (*Ornithorhynchus anatinus*), none of which was able to support replication of hmtDNA (Table 1, Figure 2). Collectively, these results indicate that while the ability of TFAM orthologs to support replication of hmtDNA is affected by the phylogenetic distance, it is not solely determined by it. Nor is it solely dictated by the percentage of consensus/identical amino acids in the mature portion of TFAM as TFAM from *X. leavis* (53.7%/37.1%) supported hmtDNA replication, whereas TFAMs from *D. novemcinctus* (77.7%/64%) and *S. harrisii* (67.8%/48.7%) did not (Table 1).

### 3.5. Coelacanth TFAM Promotes hmtDNA Instability

The ability of TFAM from such an ancient vertebrate as coelacanth to support hmtDNA replication prompted us to take a closer look at cells expressing this oTFAM. In these cells, hmtDNA was maintained at reduced mtCN as compared to cells expressing hTFAM, and upon continuous propagation in +UP media, mtCN in these cells was further reduced. (Figure 3). These cells also did not grow well in −UP media, which suggested that some of the cells in the population may have lost mtDNA. However, cells that survived in -UP media had mtCNs only slightly reduced compared to cells expressing hTFAM, and that deficiency could be rescued by transducing cells with a retrovirus encoding hTFAM (rv.5132, Appendix A). Consistent with the notion of hmtDNA instability in cells expressing *coelTFAM*, when these cells adapted to -UP media were grown for 5 weeks in +UP media and then cloned, some of the resulting clones lacked mtDNA (Figure 3). Moreover, some of the clones demonstrated evidence of reduced mtDNA content, which was confirmed by dddPCR (Figure 3C,D). Cells expressing coelTFAM were significantly more glycolytic, and this defect was also rescued by transducing them with *hTFAM*. The rescue of OXPHOS and expression of mtDNA-encoded polypeptides did not correlate well with steady-state levels of mitochondrial mRNAs but correlated with steady-state levels of MT-RNR2 (Figure 3).

### 3.6. oTFAM MTSs Are Functional in Human Cells

The inability of oTFAMs to support hmtDNA replication, conceivably, could be mediated by a failure of corresponding MTSs to target oTFAMs to the mitochondrial matrix in human cells. Therefore, we tested the functionality of oTFAM MTSs in human cells by testing their ability to direct EGFP to mitochondria in 143B cells. Generally, oTFAM MTSs were functional in human cells, with very few exceptions (Appendix A). A fusion between Tasmanian devil TFAM MTS and EGFP failed to express. However, in the course of this study, the original Tasmanian devil TFAM entry (XP_003755126) has been updated with a different MTS (XP_031812818), which is functional in human cells (Appendix A). Similarly, the TFAM sequence for green sea turtle (XP_007060730) also was updated with a new MTS (XP_037761268). Apart from these two, only MTSs from elephant shark and bald eagle were unable to direct EGFP to mitochondria in human cells, and MTS from platypus was partially functional (Appendix A). However, the shark-human chimeric TFAM was functional (Figure 4), suggesting that even though shark TFAM MTS was unable to deliver EGFP, it successfully targeted the chimeric TFAM.

### 3.7. TFAM C- and N-Terminal Domains Are Functionally Independent

To resolve the failure of some oTFAMs to support hmtDNA replication with greater granularity, we implemented a domain-swapping approach by combining hTFAM and oTFAM NTDs and CTDs so that the resulting chimeras possess one domain from hTFAM and another domain from oTFAM (Figure 1 and Figure 4, and Appendix A). In eight instances, both NTD and CTD of oTFAMs supported hmtDNA replication as parts of chimeras. In alligator, acorn worm, and nematode, neither NTD nor CTD was functional in chimeras with hTFAM. Finally, in lancelet, *Drosophila*, and sea urchin, only NTD was functional in chimeras, thus suggesting that CTD may play the leading role in determining the species-specificity of oTFAMs (Figure 4).

We also examined alignments of the functional oTFAMs and chimeras to identify conditionally permissive substitutions. The condition here being the context of other substitutions present in the same variant TFAM. Surprisingly, even in our limited set, only 23 aa in the mature hTFAM were invariant. Of those, none were in the leader sequence, six invariant aa were in HMG1, 14 were in HMG2, and three in the tail (Figure 5). HMG2 constitutes 32% of the mature hTFAM length, yet it hosts 61% of residues potentially critical for hmtDNA replication. This is also consistent with the notion that the CTD domain plays the leading role in determining the species-specificity of oTFAMs. Overall, we detected 730 conditionally permissive substitutions in 204 aa mature form of hTFAM (Figure 5).

## 4. Discussion

Here, we describe the first “clean” reverse genetic technique for the analysis of proteins involved in mtDNA replication, the GeneSwap approach. Unlike some other methods that have been or potentially can be used for this purpose the GeneSwap approach is either faster, cleaner (no wt GOI co-expression—compare to, e.g., [8,25,44]), or both.

In this study we demonstrated that the lethality of the knockouts of the critical components of the mtDNA replication apparatus is conditional. Independently, D. Kang’s group reached the same conclusion for a subset of proteins examined in this study [12,13]. These observations open opportunities for reverse genetic analysis of proteins involved in mtDNA replication. It also conclusively resolves the argument whether TFAM or any other tested protein in this study plays an essential nuclear role.

In vitro, transcription from mitochondrial promoters can be initiated in the presence of only two proteins, mitochondrial RNA polymerase (POLRMT) and mitochondrial transcription factor B2 (TFB2M). This observation led to the model that describes the core human mitochondrial transcription apparatus as a regulated two-component system [45,46,47]. This model is supported by in vivo studies, in which mitochondrial transcription is reduced but not lost in *TFAM* KO tissues [48,49]. However, this model is not universally accepted, and an alternative model argues that the experimental conditions under which two-component transcription takes place may not faithfully recapitulate those found in vivo because they favor promoter “breathing” (i.e., spontaneous separation of DNA strands) and that TFAM is also a component of the core mitochondrial transcription apparatus [50]. Our observations indicate that in the absence of TFAM, mtDNA (and, therefore, mitochondrial transcription) is lost. Hence, our observations suggest that in situ, the core mitochondrial transcription apparatus is an obligate three-component system.

Currently, it remains unresolved whether human and mouse mtDNA contains two (HSP and LSP) or three (HSP1, HSP2, and LSP) promoters [51]. Both in vivo and in vitro data support the existence of the two heavy-strand transcription initiation sites in both human and murine cells [47,52,53,54]. Moreover, in vitro MTERF1 stimulates transcription initiation at only one of the HSP start sites [55], thus supporting the two-HSP-promoters model. However, the close proximity of the two major HSP initiation sites in murine mtDNA and MTERF1 knockout data were interpreted in favor of the single-HSP model [56,57]. One limitation of our approach as applied to mitochondrial transcription is that we measure the steady-state levels of mitochondrial transcripts rather than initial transcription rates. The steady-state levels of mitochondrial transcripts are affected by both transcription rates and transcript stability. However, in the absence of evidence that TFAM is involved in the regulation of mitochondrial transcript stability, the steady-state levels of transcripts in situ may be considered a reasonable surrogate of transcription rates. With this in mind, the most parsimonious explanation of our observations of differential effects of the coelTFAM on steady-state levels of the MT-RNR2 and MT- ND1/MT-CO1 is that there are two heavy-strand promoters in situ.

One conclusion of this study is that TFAM NTD and CTD can function independently from each other in mtDNA replication. This independence of TFAM domains allowed us to generate functional chimeras. This observation may be useful in future studies aimed at engineering TFAMs with customized properties tuned for particular applications. Aside from the studies on mtDNA replication we can also envision the utility of the GeneSwap approach described here in studies on mtDNA transcription and in engineering cell lines with modified mtDNA-related functions for elucidating the role of these functions in (patho)physiological processes.

Another interesting observation made in this study is that in those cases where it could be resolved, the inability of TFAM orthologs to support replication of hmtDNA is not attributable to any specific aa substitution but instead represents a cumulative effect of many mutations. In our limited set of oTFAMS and their domains, which were either wholly or partially (one domain) functional in hmtDNA replication, only 23 aa were invariant, and a total of 730 different substitutions were observed in the 204 aa mature form of hTFAM. Therefore, TFAM appears to be remarkably tolerant to aa substitutions. In contrast, insertions and deletions in oTFAM HMG domains were not tolerated. Furthermore, adjusting oTFAM length to that characteristic of hTFAM (e.g., by deleting the three extra aa from the *Drosophila* HMG2, or five aa from HMG2 of sea urchin, or inserting one “missing” aa in lancelet’s HMG2 (Appendix A)) did not render these proteins competent in hmtDNA replication (results not shown).

mtDNA is segregationally stable, and to our knowledge, spontaneous loss of mtDNA had not been previously reported. Here we show that coelTFAM induces a moderate segregational instability of mtDNA in human cells. Cells exhibiting such instability represent a convenient in situ model to study mtDNA segregation and copy number control.

On all three occasions when only one oTFAM domain supported hmtDNA replication, that was NTD. This, combined with the partitioning of the 61% of invariant aa residues to HMG2, provides strong support for the notion that HMG2 plays the leading role in shaping TFAM species-specificity.

The genome of armadillo, *D. novemcinctus*, encodes two hTFAM orthologs (Table 1). One of those, XP_004473261, supported hmtDNA replication, while XP_004473258 did not. Of note, XP_004473258 has a one aa deletion in the linker region. Therefore, the functionality of the arTFAM-hTFAM chimera, which also contains this deletion, suggests that the hTFAM may be tolerant to the linker length variability.

In this study, we observed that, with two exceptions, oTFAM MTSs were functional in human cells. This observation suggests remarkable evolutionary conservation of mitochondrial protein import. Inaccuracies of computational gene annotations can plausibly account for the two observed exceptions. Indeed, during the span of this project, MTS in the Tasmanian devil TFAM was revised, and in the corrected sequence MTS was functional in human cells. It has been previously reported that some MTSs efficiently target to mitochondria their respective cognate proteins, but not a fluorescent reporter, which may provide another plausible explanation for exceptions [58].

To effect mtDNA replication and transcription, TFAM interacts with both mtDNA and other proteins. The GeneSwap approach opens opportunities to study these complex relationships in situ using reverse genetics. It would be of interest to identify the structural and biochemical basis behind the inability of some oTFAMs to support hmtDNA replication in situ. In this respect, reviewing the differences in mtDNA organization between taxonomical groups can be instructive. For example, tRNA genes around OriL in mitochondrial genomes of opossum and Tasmanian devil are rearranged. tRNA genes create an extensive secondary structure around OriL, which can be potentially recognized by TFAM. This raises a provocative question of whether opossum and Tasmanian devil TFAMs fail to support hmtDNA replication because of their inability to properly recognize/position at OriL. In a similar vein, in the acorn worm, mtDNA organization is dramatically different from hmtDNA. Apart from the different gene order, the control region is inserted between MT-ATP8 and MT-ATP6 genes (these two genes overlap in hmtDNA), not flanked by tRNA genes, and is located approximately 1/3 genome away from rRNA genes. Correspondingly, acorn worm TFAM has a four aa deletion in HMG1, a six aa deletion in the linker, and a one aa insertion in the HMG2 domain vs. hTFAM. It remains unclear which of these differences in TFAMs represent adaptations to structural changes in mtDNA. However, we believe that the GeneSwap approach described here will become an important tool in addressing these questions related to TFAM-mtDNA coevolution as well as other questions related to mtDNA replication.

## Figures and Tables

**Figure 1 cells-11-02168-f001:**
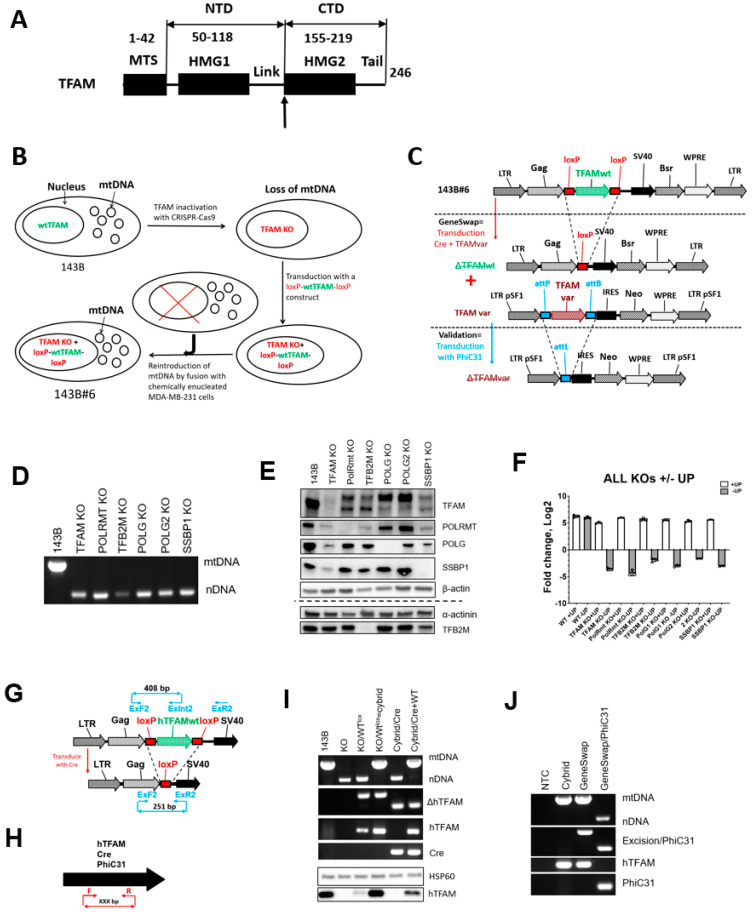
The domain structure of the human TFAM and a general outline of the GeneSwap approach. (**A**), Upward vertical arrow, a crossover point in chimeras. Here, we refer to the TFAM portion preceding the crossover point as N-terminal domain (NTD), and the portion following HMG2 as C-terminal domain (CTD). Domain boundaries are given in aa coordinates. MTS, matrix targeting sequence, Link, linker region. (**B**), Engineering of a GeneSwap cell line using *TFAM* GeneSwap 143B#6 as an example. The three main steps include: (1) A gene of interest (*TFAM*) is inactivated by CRISPR-Cas9, (2) the resulting ρ0 cells are transduced with a retrovirus encoding a wt *hTFAM* gene “floxed” with loxP sites for Cre recombinase, (3) mtDNA is reintroduced into these transduced ρ^0^ cells by fusing them with enucleated cells. (**C**), To implement the GeneSwap approach, 143B#6 cells (the top construct) are co-transduced with retroviruses encoding Cre recombinase and altered *TFAM* (TFAMvar, central panel.). This results in the excision of the *wtTFAM* and simultaneous re-expression of the *TFAMvar*. In the resulting co-transductants, mtDNA is retained only if TFAMvar is functional in hmtDNA replication. In that case the functionality is further validated by transducing cells with PhiC31 recombinase, which effects the loss of *TFAMvar* and hmtDNA (the bottom construct). (**D**), 143B cells KO for *hTFAM*, *hPolRmt*, *hTFB2M, hPolG1, hPolG2*, and *hSSBP1*, are viable, but lose hmtDNA. (**E**), Validation of the *hTFAM, hPolRmt, hTFB2M, hPolG1*, and *hSSBP1* KO by Western blotting. (**F**), 143B cells KO for for *hTFAM, hPolRmt, hTFB2M, hPolG1, hPolG2*, and *hSSBP1* survive in +UP media, but die in −UP media. A representative of two independent experiments, each with three technical replicates. (**G**), A diagram for PCR-genotyping of *ΔhTFAM* (panel (**I**)). (**H**) A diagram for PCR genotyping of *hTFAM, Cre*, and *PhiC31* (panels (**I**,**J)**). Primer sequences and fragment sizes as in Appendix A. (**I**) A PCR genotyping and Western blotting verification of the main steps in the engineering and validation of the 143B#6 *TFAM* GeneSwap cell line. KO, CRISPR-Cas9 *hTFAM* KO; KO/WTlox, *TFAM* KO cells transduced with rv.4000 encoding wt *hTFAM*; KO/WTlox = cybrid, reintroduction of hmtDNA in KO cells complemented with rv.4000; Cybrid/Cre, Cre/lox deletion of the wt *hTFAM* introduced with rv.4000; GeneSwap, GeneSwap of the wt *hTFAM* encoded by rv.4000 for wt *hTFAM* encoded by rv.5460. HSP60, loading control for Western blotting. PCR subpanels (top to bottom): Top, duplex PCR for nDNA and mtDNA. ΔhTFAM, diagnostics for excision of *hTFAM* encoded by rv.4000, see diagram in (**G**); hTFAM, detection of the cDNA for wt *hTFAM* encoded by rv.4000 and rv.5460, see diagram in (**H**); Cre, detection of *Cre* recombinase gene encoded by rv.3442, see diagram in (**H**). (**J**) GeneSwap/PhiC31, PhiC31-mediated excision of the wt *hTFAM* encoded by rv.5460 is accompanied by the loss of mtDNA. nDNA, mtDNA, duplex PCR for nDNA, and mtDNA. Note that due to high copy number, mtDNA suppresses amplification of the nDNA. NTC, no template control. Subpanels as in (**I**) plus: Excision/PhiC31-PCR diagnostic of *hTFAM* excision from rv.5460, see the middle and the bottom of panel C; PhiC31, PCR detection of *PhiC31* gene, see diagram in (**H**). See Appendix A for primer sequences and predicted fragment sizes.

**Figure 2 cells-11-02168-f002:**
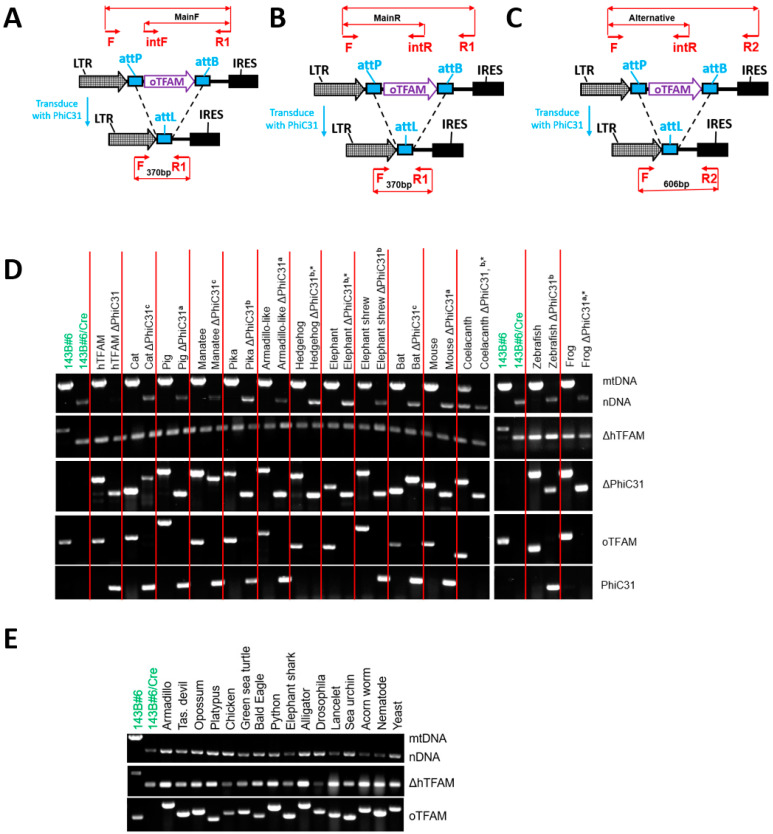
Genotyping cells transduced with *TFAM* orthologs. (**A**–**C**) The three alternative strategies for geno-typing excision of *oTFAM* (PCR subpanel ΔPhiC31 in (**D**)): The Main strategy with forward internal primer (**A**), the Main with reverse internal primer (**B**), and Alternative strategy (**C**). (**D**), PCR-genotyping of the clones in which hmtDNA replication is supported by *oTFAM*s. Superscripts a, b, and c correspond to genotyping strategies in (**A**–**C**). (**E**) PCR genotyping of cells expressing *oTFAM*s that are unable to support hmtDNA replication. Geno-typing for mtDNA, nDNA, *ΔhTFAM*, and *PhiC31* are essentially as depicted in Figure 1I,J. *oTFAM*s were geno-typed as diagramed in Figure 1H. * These orthologs were excised by transiently transfecting cells with pMA4854 (Appendix A). See Appendix A for primer sequences and predicted fragment sizes.

**Figure 3 cells-11-02168-f003:**
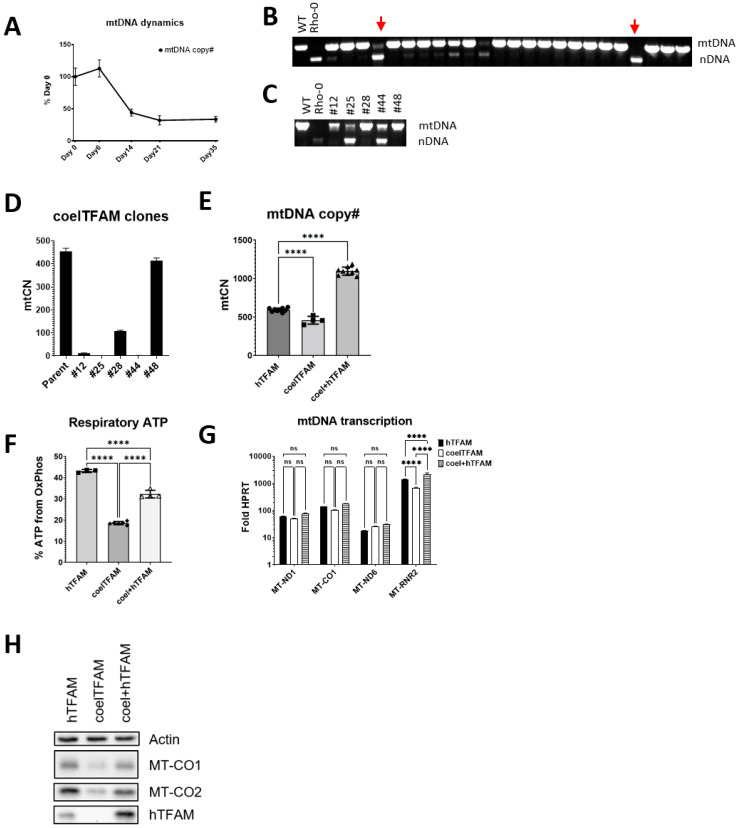
Coelacanth TFAM promotes spontaneous loss of hmtDNA. (**A**), Upon cultivation in +UP media, mtCN in cells expressing coelTFAM is reduced. Cells expressing coelTFAM were incubated in +UP media for up to 35 days, cell aliquots were collected at indicated intervals, and mtCN was determined by qPCR. (**B**), coelTFAM promotes spontaneous loss of mtDNA. Cells were grown in +UP media for 5 weeks, plated by limiting dilution, the resulting colonies were picked and tested for the presence of mtDNA. Red vertical arrows indicate clones that lost mtDNA. (**C**,**D**), in the experiment described in (**B**), clones representative of various mtCNs were analyzed by conventional (**C**) and dddPCR. (**D**). (**E**), Transduction with *hTFAM* augments mtCN in cells expressing coelTFAM. Cells expressing coelTFAM were selected in –UP media and transduced with a retrovirus rv.5132 (Appendix A), which expresses *hTFAM*. ****, *p* < 0.0001. (**E**), One-way ANOVA with post hoc Tukey test. A representative of two independent experiments. (**F**–**H**), Changes in OXPHOS and mtDNA-encoded OXPHOS subunits expression correlate with expression of the MT-RNR2. (**F**), OXPHOS deficiency in cells expressing coelTFAM can be partially rescued with hTFAM. ****, *p* < 0.0001. One-way ANOVA with post hoc Tukey test. A representative of two independent experiments. (**G**), coelTFAM does not significantly affect expression of the *MT-ND1, MT-CO1*, or *MT-ND6*, but affects expression of the *MT-RNR2*. Two-way ANOVA with Dunnet correction. ****, *p* < 0.0001; ns, not significant. (**H**), OXPHOS correlates with expression of mtDNA-encoded subunits MT-CO1 and MT-CO2. A representative of two independent experiments.

**Figure 4 cells-11-02168-f004:**
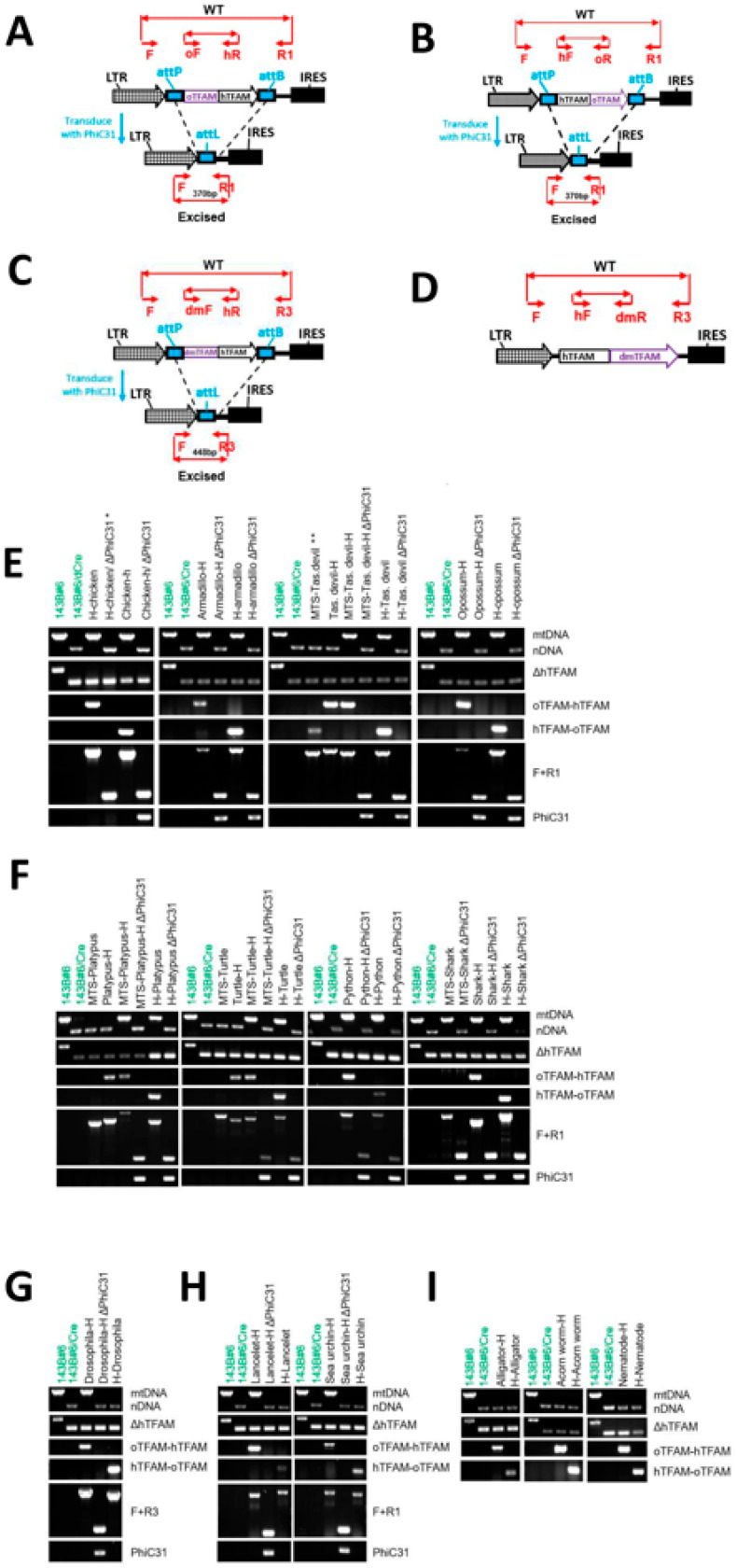
Screening oTFAM chimeras for their ability to support replication of hmtDNA. (**A**,**B**), PCR genotyping strategies for cells expressing *oTFAM* chimeras. (**C**,**D**), PCR genotyping strategies for cells expressing *dmTFAM-hTFAM* and *hTFAM-dmTFAM* chimeras, respectively. (**E**,**F**), PCR genotyping of *oTFAM* chimeras in which both NTD and CTD support hmtDNA replication (**G**,**H**), PCR genotyping of *oTFAM* chimeras in which only NTD supports hmtDNA replication (**I**), PCR genotyping of *oTFAM*s chimeras in which neither NTD, nor CTD supports hmtDNA replication. MTS, matrix targeting sequence of the human ornithine transcarbamylase was appended in front of oTFAM. *, excised by transiently transfecting cells with pMA4854 (Appendix A). **, a spurious PCR product in *hTFAM-oTFAM* PCR due to homology between *hTFAM* and *tdTFAM*. mtDNA, nDNA, *ΔTFAM*, and *PhiC31* genotyping as in Figure 1. Subpanel oTFAM-hTFAM, detection of the corresponding chimeras (see diagrams (**A**,**C**)). Subpanel hTFAM-oTFAM, detection of the corresponding chimeras (see diagrams (**B**,**D**)). Subpanels F + R1 and F + R3, detection of chimera excision (see diagrams (**A**–**D**)). See Figure 1A for chimera design and Appendix A for primer sequences and predicted amplicon sizes.

**Figure 5 cells-11-02168-f005:**
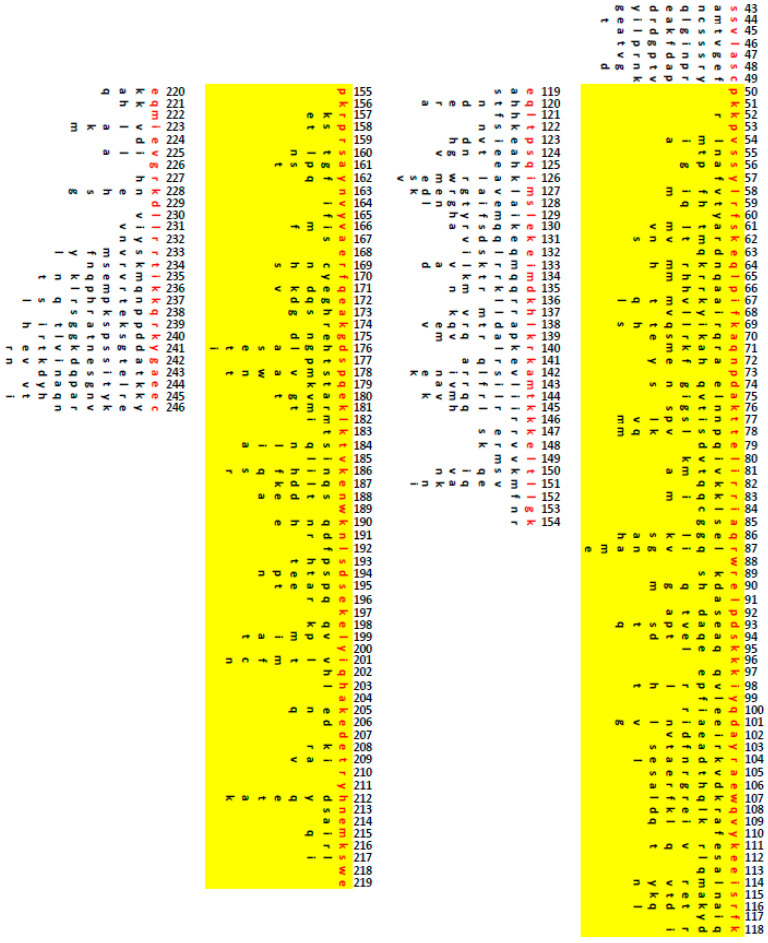
Conditionally permissive substitutions in hTFAM. The numbering corresponds to aa position in hTFAM precursor (before MTS removal). Red font, aa in wt hTFAM. Black font, conditionally permissive substitutions at each position. Yellow highlights, HMG1 and HMG2 domains.

**Table 1 cells-11-02168-t001:** Replication of the human mtDNA by TFAM orthologs.

Species, Trivial	Species Latin/GenPept	Consensus/Identity, % ^1^	Comp ^2^	Species, Trivial	Species Latin/GenPept	Consensus/Identity, % ^1^	Comp ^2^
Human	Homo sapiens/NP_003192.1	100/100	+	Chicken	Gallus gallus/NP_989431	61/42.2	−
Cat	Felis catus/XP_003993997	84.3/74.1	+	Green Sea Turtle	Chelonia mydas/XP_007060730	58.5/43.2	−
Pig	Sus scrofa/NP_001123683	84.3/71.6	+	Bald eagle	Haliaeetus leucocephalus/XP_010578303	58.4/41.6	−
Manatee	Trichechus manatus latirostris/XP_004369930	82.3/73.6	+	Python	Python bivittatus/XP_007424936	56.2/40.4	−
Pika	Ochotona princeps/XP_004583634	81.2/70.1	+	Coelacanth	Latimeria chalumnae/XP_006004778	55.9/41.5	+
Armadillo-like	Dasypus novemcinctus/XP_004473261	81.2/69.5	+	Zebrafish	Danio rerio/NP_001070857	53.7/39	+
Hedgehog	Echinops telfairi/XP_004701428	80.9/66.8	+	Frog	Xenopus leavis/NP_001081106	53.7/37.1	+
Elephant	Loxodonta Africana/XP_003409078	80.4/68.8	+	Elephant shark	Callorhinchus milii/XP_007895218	50.7/37.0	−
Elephant shrew	Elephantulus edwardii/XP_006895656	80.2/69.5	+	Alligator	Alligator sinensis/XP_006032464	50.6/38.2	−
Bat	Myotis lucifugus; XP_006098959	79.7/63.5	+	Fruit fly	Drosophila melanogaster/NP_524415	47.3/30.9	−
Armadillo *	Dasypus novemcinctus; XP_004473258	77.7/64	-	Lancelet	Branchiostoma floridae/XP_035670496	39.1/25.3	−
Mouse	Mus musculus/NP_033386.1	76.1/62.9	+	Sea urchin	Strongylocentrotus purpuratus/XP_030834910.1	39.1/24.8	−
Tasmanian devil	Sarcophilus harrisii/XP_003755126	67.8/48.7	−	Acorn worm	Saccoglossus kowalevskii/XP_006813645	38.4/25.5	−
Opossum	Monodelphis domestica/XP_007478397	65.8/48.7	−	Nematode	Caenarhabditis elegans/NP_501245.1	38.4/21.2	−
Platypus	Ornithorhynchus anatinus XP_001507982	65/44.7	−	Yeast	Saccharomyces cerevisiae/NP_013788.1	27.2/17.4	−

^1^ For mature TFAMs. Here, defined as beginning with the first aa residue in the HMG1 domain. ^2^ The ability to complement hTFAM deficiency (support hmtDNA replication). **+**/**−**, able/unable, respectively. * The linker domain of this ortholog is shorter by 1 aa.

## Data Availability

The data presented in this study are contained in the article and Appendix A.

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
