# Peer review of "A Method for In Situ Reverse Genetic Analysis of Proteins Involved mtDNA Replication"

_cells, 2022, doi:10.3390/cells11142168_

Round 1

Reviewer 1 Report

The manuscript, entitled “A method for in situ reverse genetic analysis of proteins involved mtDNA replication”, describes a very interesting method that can significantly expand the possibilities of studying the replication and maintenance of the mitochondrial genome. The authors used the CRISPR-Cas9 system to create an experimental setup that has not been available for years in human cell line research. The GeneSwap approach developed by them enables the elegant inactivation of the gene encoded in nuclear DNA with the use of an appropriately designed sgRNA, then carrying out rescue experiments using retroviral vectors, in which the gene of interest is flanked by sequences recognized by Cre or PhiC31 recombinase. This enables the gene to be excised in order to verify phenotype reversal. Kozhukhar et al. choose TFAM protein as an object for detailed testing of their method, which additionally leads to some interesting conclusions regarding the role of this protein in the functioning of the mitochondrial genetic apparatus.

The GeneSwap method presented here is an interesting proposal, for studies on the pathogenesis of mitochondrial diseases, in which mutations in nuclear-encoded genes responsible for replication and maintenance of the mitochondrial genome that lead to the appearance of mutations in the mitochondrial DNA. This method would allow verification whether a mutation in the gene encoding a given protein actually leads to changes in mtDNA.

In conclusion, the manuscript has a high scientific and cognitive value and should be published in Cells, but before that, several of the following issues need clarification.

  1. First of all, the text of the manuscript requires intensive editing.

    1. Most of the legend in Figure 3 is missing.

    2. One part of the legend in Figure 1 is in the main text.

    3. The vector maps are present in Figure S1 and not in Figure S2, many references in the main body of the manuscript should be changed accordingly.

    4. Figure S2 applies to STAR Methods (not relevant for MDPI Journals) and the Supplementary figure S2, should be changed accordingly.

    5. Line 112: Sanger sequencing (Figure S1) should be Figure S2

    6. Line 179-180: Supplemental Figures do not include such schemes

    7. Line 295: Figure S1 should be changed to Figure S2

    8. Line 412: Figure S2 and Table S2 should be changed to the appropriate references

    9. Figure 4 legend requires a spell check

  2. The anti-TFAM western blotting result in Figure 1E looks strange, the lane for untreated cells differs significantly from the rest. I suggest showing an even larger area of the membrane so that it is clear that also for untreated cells two bands are visible. What could be the reason for such a difference in TFAM migration in untreated 143B compared to other experimental conditions? The image in the TFAMKO lane suggests that both observed bands represent TFAM, how could these protein isoforms differ?

  3. The authors should refer in the text to the relationships between the levels of individual tested proteins in KO lines, as shown in Figure 1E.

  4. How do the authors explain such a significant increase in the number of mtDNA copies in coel+hTFAM cells (Figure 3E)?

  5. The authors should explain in the Materials and Methods section why they use MDA-MB-231 cells to reintroduce mtDNA. Do the mtDNAs of these cells and 143B cells belong to the same haplogroup? In the light of recent studies, this is not without significance for the metabolic functions of mitochondria and may have an impact on the efficiency of rescue experiments, especially in the context of studying the functions of OXPHOS.

  6. The authors should discuss the advantages of the GeneSwap approach compared to other available methods of functional analysis of human proteins, for example, based on vectors with a bidirectional promoter and silencing of expression using miRNA, such as described in Szczesny et al., 2018 (PMID: 29590189).

Author Response

We are grateful to Reviewer#1 for their insightful comments, suggestions, and queries, which were addressed as described below.

Reviewer#1 comments:

The manuscript, entitled “A method for in situ reverse genetic analysis of proteins involved mtDNA replication”, describes a very interesting method that can significantly expand the possibilities of studying the replication and maintenance of the mitochondrial genome. The authors used the CRISPR-Cas9 system to create an experimental setup that has not been available for years in human cell line research. The GeneSwap approach developed by them enables the elegant inactivation of the gene encoded in nuclear DNA with the use of an appropriately designed sgRNA, then carrying out rescue experiments using retroviral vectors, in which the gene of interest is flanked by sequences recognized by Cre or PhiC31 recombinase. This enables the gene to be excised in order to verify phenotype reversal. Kozhukhar et al. choose TFAM protein as an object for detailed testing of their method, which additionally leads to some interesting conclusions regarding the role of this protein in the functioning of the mitochondrial genetic apparatus.

The GeneSwap method presented here is an interesting proposal, for studies on the pathogenesis of mitochondrial diseases, in which mutations in nuclear-encoded genes responsible for replication and maintenance of the mitochondrial genome that lead to the appearance of mutations in the mitochondrial DNA. This method would allow verification whether a mutation in the gene encoding a given protein actually leads to changes in mtDNA.

In conclusion, the manuscript has a high scientific and cognitive value and should be published in Cells, but before that, several of the following issues need clarification.

  1. First of all, the text of the manuscript requires intensive editing.

    1. Most of the legend in Figure 3 is missing.

The rest of the Figure 3 legend has been added

    1. One part of the legend in Figure 1 is in the main text.

It only appears that way because the second part of the Figure 1 legend was transferred to the next page

    1. The vector maps are present in Figure S1 and not in Figure S2, many references in the main body of the manuscript should be changed accordingly.

Indeed. References have been changed accordingly.

    1. Figure S2 applies to STAR Methods (not relevant for MDPI Journals) and the Supplementary figure S2, should be changed accordingly.

Actually, reference to STAR methods was in the Figure 1 legend. It has been removed.

    1. Line 112: Sanger sequencing (Figure S1) should be Figure S2

Changed as per comment c.

    1. Line 179-180: Supplemental Figures do not include such schemes

Changed to Figure 1G

    1. Line 295: Figure S1 should be changed to Figure S2

Changed as per comment c.

    1. Line 412: Figure S2 and Table S2 should be changed to the appropriate references

Changed as suggested

    1. Figure 4 legend requires a spell check

Corrected

  1. The anti-TFAM western blotting result in Figure 1E looks strange, the lane for untreated cells differs significantly from the rest. I suggest showing an even larger area of the membrane so that it is clear that also for untreated cells two bands are visible. What could be the reason for such a difference in TFAM migration in untreated 143B compared to other experimental conditions? The image in the TFAMKO lane suggests that both observed bands represent TFAM, how could these protein isoforms differ?

Full uncropped original gel and membrane images were submitted for review with the manuscript as per instructions to the authors and can be examined. We suggest that, unlike wt 143B cells, rho-0 cells with KO in proteins involved in mtDNA replication may have impaired mitochondrial import of TFAM. Therefore, the band with a larger molecular weight that appears in these KO cells may represent an unprocessed (cytosolic?) TFAM (TFAM + MTS). This phenomenon has been recently described for POLRMT and TFB2M KO cells by Inatomi et al. (ref. 12).

  1. The authors should refer in the text to the relationships between the levels of individual tested proteins in KO lines, as shown in Figure 1E.

We do not understand the relevance of the levels of individual tested proteins in Figure 1E to the GeneSwap technique or its validation. Figure 1E is meant to demonstrate the absence of proteins in KO cells. We believe that other comparisons as suggested by the reviewer are immaterial to the point that we are making in this panel.

  1. How do the authors explain such a significant increase in the number of mtDNA copies in coel+hTFAM cells (Figure 3E)?

This is a consistent observation for which we do not have a good explanation yet, but actively pursuing.

  1. The authors should explain in the Materials and Methods section why they use MDA-MB-231 cells to reintroduce mtDNA. Do the mtDNAs of these cells and 143B cells belong to the same haplogroup? In the light of recent studies, this is not without significance for the metabolic functions of mitochondria and may have an impact on the efficiency of rescue experiments, especially in the context of studying the functions of OXPHOS.

MDA-MB-231 cells were chosen as a donor to enable differentiation of true cybrids from donor cells that escaped mitotic inactivation by STR profiling. The Materials and Methods section has been amended accordingly. We do not know the haplogroup of MDA-MB-231 cells. However, we do agree that no direct comparisons of OXPHOS should be made between cell lines with different mtDNA haplogroups. Therefore, in our experiments, mtDNA haplogroup stays the same (MDA-MB-231), and the only direct variable is the number of aa substitutions in oTFAMs vs. hTFAM. Hence, we believe that our attribution of observed changes in OXPHOS to the differences in TFAM sequence in this study is justified.

  1. The authors should discuss the advantages of the GeneSwap approach compared to other available methods of functional analysis of human proteins, for example, based on vectors with a bidirectional promoter and silencing of expression using miRNA, such as described in Szczesny et al., 2018 (PMID: 29590189).

A discussion has been added.

Reviewer 2 Report

Manuscript entitled with “A method for in situ reverse genetic analysis of proteins involved mtDNA replication” by Kozhukhar et al. Authors used CRISPR-Cas9 mediated gene editing technique for tissue specific knockout of key protein involved in mtDNA replication. Reverse genetic analysis (GeneSwap) can be used for the study of mtDNA replication and TFAM-mtDNA coevolution related studies. Authors show the lethality of the knockout of the critical component of mtDNA replication apparatus is conditional. Study is also supported independently by D. Kangs group. Authors nicely designed the study and executed the experiments. Overall data presented in the manuscript are well planned. Writing part of the manuscript is good and discussed in detail. I would like to recommend the manuscript accept in its current form.   

Author Response

We thank the Reviewer for his/her kind comments